# Cascaded Contrastive Medical Language-Image Pretraining on Medical Images

## Abstract

Due to the concise design and the wonderful generalization performance, contrastive language-image pre-training (CLIP) has been investigated in the medical domain for medical image understanding. However, few studies have been done on CLIP for multilevel medical information alignment. In this paper, we proposed cascaded CLIP (casCLIP) where contrastive alignment is performed on multilevel information. In addition, we propose aligning the report with the entire image series and employ a multi-layer transformer to integrate the image embeddings from a study into a single embedding of image series. Moreover, we introduce support alignment opposition de-alignment method to enhance higher-level alignment. In this study, casCLIP was pre-trained on a dataset of chest X-ray images with reports and the high level disease information extracted from the reports. Experimental results on multiple public benchmarks demonstrate the effectiveness of our model for zero-shot classification.

## 1 Introduction

Contrastive language-image pre-training (CLIP) (Radford et al., 2021) is a cutting-edge model renowned for its powerful function in associating images and text. It is well appreciated for its elegant design and exceptional ability to generalize across various domains, making it a valuable tool for bridging the gap between visual data and language. In the medical domain, interpreting medical images, such as X-rays, MRIs, and CT scans, is critical for diagnosing diseases and developing treatment plans. The potential of CLIP in medical image understanding lies in its capacity to associate medical images and reports making it garner much attention in the context of the medical field recently, such as MedCLIP (Wang et al., 2022), CheXzero (Zhang et al., 2023), MedKlip (Wu et al., 2023), GloRIA (Huang et al., 2021a).

Medical data usually contains hierarchical labels that provide different levels of disease information. However, existing CLIP methods were not explicitly designed to handle this multilevel medical information. To illustrate this, consider a study involving medical images and their corresponding original reports. From this data, we can derive a level 1 summary of the presented diseases, such as atelectasis. This disease, in turn, falls under the category of chest diseases, representing level 2 information. Each level provides a distinct level of abstract description for the images. Conventional CLIP methods attempt to align a specific image embedding with these text embeddings corresponding to the multi-level information (Figure 1a), which may result in underrepresentation of the embeddings. Therefore, there is a significant gap in developing a comprehensive approach that integrates multilevel information for more accurate medical image understanding and diagnosis.

To fill this gap, we propose cascaded CLIP (casCLIP), which employs a cascading mechanism to facilitate multilevel information alignment. As illustrated in Figure 1b, besides the alignment between image embedding from the image encoder and the embedding of the original report, casCLIP also aligns the embeddings of higher level text summaries with higher level image embeddings that are derived from the lower level image embeddings. The goal of casCLIP is to create multilevel representations that can effectively capture the diverse levels of information present in both images and their associated text, ultimately leading to more precise image and text representations.

Furthermore, it is important to note that a medical report often encompasses a series of medical images within a single study. While the report can provide an overview of the entire study of image series, it may not necessarily correspond to each individual image within it. For example, consider

(a) conventional CLIP        (b) casCLIP

Figure 1: The illustration of handling multi-level text information for conventional CLIP and casCLIP. (a) Conventional CLIP aligns a specific image embedding with multiple text embeddings corresponding to the multi-level information; (b) casCLIP generates multiple image embeddings for the multiple text levels and cascades multiple alignments between text embeddings and image embeddings. In the high-level alignment process, casCLIP also de-align the opposite text summary to the image embedding.

a chest radiograph study consisting of frontal and lateral images. Pneumonia might be detectable in the frontal image but not in the lateral one. Therefore, attempting to pair the report with each individual image is not suitable. While most previous research has focused on associating the report with each image separately for contrastive learning, we propose aligning the report with the entire image series study. In this paper, we treat the image series within a study as an integrated entity and employ a multi-layer transformer (Vaswani et al., 2017) to integrate the image embeddings from the study into a single image series embedding, which can then be aligned with the report embedding.

To achieve better alignment in the high-level alignment process, we employ a technique referred to as Support Alignment Opposition De-alignment (SAOD). This approach involves creating an opposite text summary and increasing the dissimilarity between the opposite text summary and the image, while simultaneously bringing the support text summary closer to the image in terms of similarity. As depicted in Figure 1b, for the level 1 summary "*Disease atelectasis is found*", we construct a corresponding opposite text summary "*Disease atelectasis is not found*". The goal is to ensure that an image that aligns well with *Disease atelectasis is found* should not align with *Disease atelectasis is not found*.

We pre-trained casCLIP on MIMIC-CXR (Johnson et al., 2019), a large public dataset of chest radiographs with free-text radiology reports. We evaluated the pre-trained model rigorously on numerous public benchmarks. Experimental results showed that our model achieves better performance compared to other methods in previous works. Our contributions are summarized as follows:

- **casCLIP for Multilevel Alignment:** We introduce casCLIP, a novel approach that enhances multilevel information alignment within CLIP models. This improvement leads to superior performance in downstream medical image understanding tasks.

- **Image Series Encoding:** To better model real-world medical image studies, we propose encoding the entire image series rather than individual images to align with the text. We employ a multi-layer transformer to consolidate multiple image embeddings into image series embeddings, resulting in more comprehensive representations.

- **Support Alignment Opposition De-alignment Method:** To improve alignment between high-level text summaries and their corresponding images, we introduce the SAOD alignment. This technique involves constructing opposing text summaries and intentionally increasing the dissimilarity between these opposites and the image embeddings.

- **Pre-training and Evaluation:** We conduct the pre-training of casCLIP on a large public dataset and evaluate its performance across multiple datasets. Our experimental results demonstrate the effectiveness of casCLIP in a range of medical image understanding tasks, highlighting its practical applicability.

## 2 RELATED WORK

### 2.1 VISION-LANGUAGE REPRESENTATION

Vision-language representation has undergone remarkable growth and diversification in recent research. Two prevalent architectural paradigms, the dual-stream (Jia et al., 2021; Li et al., 2021) and single-stream methods (Chen et al., 2020), have emerged as prominent strategies for fusing visual and textual modalities. Notably, several studies have endeavored to enrich vision-language models with commonsense knowledge to enhance contextual understanding and reasoning capabilities (Cui et al., 2021; Li et al., 2020; Yu et al., 2021). Additionally, there has been substantial progress in refining pretraining objectives,such as masked language modeling (MLM) and masked visual modeling (MVM) (Huang et al., 2021b; Wang et al., 2023). The CLIP family models have garnered significant attention (Radford et al., 2021; Li et al., 2022b; Mu et al., 2022; Yao et al., 2022; Chen et al., 2023), relying on vision-language contrastive learning and large-scale image-text pairs. Other CLIP models tried to explore the hierarchical information contained in data (Geng et al., 2023; Ge et al., 2023). These models have showcased impressive capabilities in aligning vision and language representations effectively. Together, this vibrant landscape of research in vision-language representation empowers a diverse array of applications, spanning from image captioning and visual question answering to specialized domains like medical imaging and diagnosis.

### 2.2 MEDICAL IMAGE UNDERSTANDING

Medical image understanding is a pivotal area in healthcare, and extensive research has been devoted to advancing the capabilities of computer systems in this domain. The field of medical image analysis has seen a surge in deep learning-based approaches, such as convolutional neural networks (CNNs), graph neural networks (GNN) and transformers, which have demonstrated remarkable success in tasks such as disease diagnosis, lesion detection, and organ segmentation (Wang et al., 2017; Mao et al., 2018; 2022). Recently, the CLIP model has sparked significant interest and represents a notable advancement. CLIP, originally designed for general vision-language tasks, has demonstrated its versatility and effectiveness in various domains, including medicine. Several adaptations and extensions of the CLIP model have been explored within the medical domain. MedCLIP (Wang et al., 2022) was designed to decouple images and texts for multimodal contrastive learning thus scaling the usable training data in a combinatorial magnitude with low cost. Huang et al. (2021a) proposed an attention-based framework for learning global and local representations for medical image recognition. Wu et al. (2023) incorporate domain-specific knowledge to CLIP to enhance medical visual-language pre-training. Zhang et al. (2023) also proposed an approach to leverage existing medical domain knowledge to guide vision-language pre-training using paired chest X-rays and radiology reports. However, few studies have been done on CLIP to handle multilevel medical information. In this paper, we propose a framework for multilevel information alignment based on contrastive learning.

## 3 METHOD

An overview of our casCLIP framework is illustrated in Figure 2. The input for casCLIP pre-training consists of three parts: the image series, the original text and higher-level text summaries. The prompts of hierarchical labels serve as the higher-level text summaries in our pre-training for medical images. Consequently, an input sample for pre-training is represented as a triplet denoted as $\mathbf{X} = (images, text, labels)$. Figure 2 illustrates the workflow of casCLIP.

### 3.1 ENCODING

**Image series encoding:** The image series encoder is to encode the image series in a study into a image series embedding. In our framework, the image series encoder consists of an image encoder and a multi-layer transformer. The image encoder is responsible for encoding each individual image within the series into an image embedding. The transformer is to consolidate these individual image embeddings into a unified image series embedding by introducing a special input [CLS], similar to the aapproach used in BERT model (Devlin et al., 2019). A projection head is applied to transform

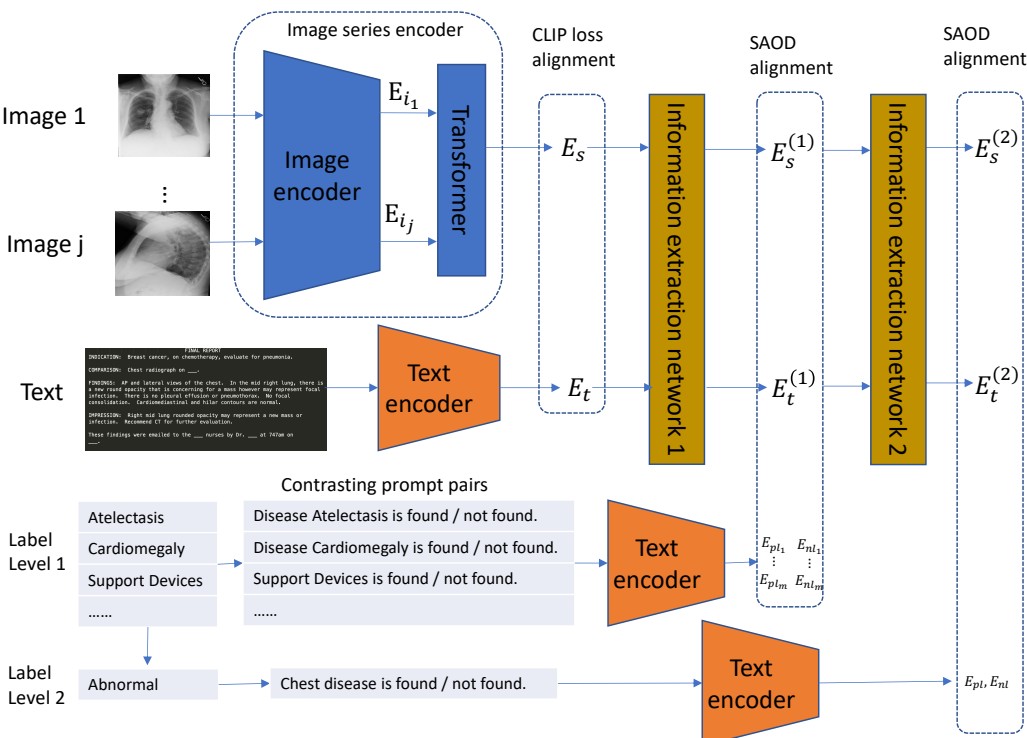

Figure 2: An overview of our framework. In the pre-training process, for a batch of input triplets like (*Image series, Text, Hierarchical labels*), the *Image series* is input to an image series encoder which consists of an image encoder and a multi-layer transformer to get an image series embedding $E_s$. Concurrently, *Text* is input to a text encoder to get a text embedding $E_t$. Each label in the *Hierarchical labels* involved in the batch is converted to multiple prompts opposite to each other, e.g., *disease found, not found or uncertain*. Each prompt is feed to the text encoder to get a label prompt embedding. Further in the process, $E_s$ and $E_t$ are passed through an information extraction network to achieve the respective level 1 embeddings $E_s^{(1)}$ and $E_t^{(1)}$ that are subsequently input to another information extraction network to get level 2 embeddings $E_s^{(2)}$ and $E_t^{(2)}$. $E_s$ and $E_t$ are aligned using the general CLIP loss function. $E_s^{(1)}$, $E_t^{(1)}$ and level 1 label prompt embeddings are aligned using the SAOD method. Similarly, $E_s^{(2)}$, $E_t^{(2)}$ and level 2 label prompt embeddings are also aligned using SAOD.

the raw embedding to a specified dimension. The process is denoted as

$$E_{i_j} = \text{Enc}_i(\mathbf{X}_{images}[j]); E_s' = \text{Transformer}([[CLS], E_{i_1}, \cdots, E_{i_j}]); E_s = f_s(E_s') \qquad (1)$$

where $\text{Enc}_i$ is the image encoder; $\mathbf{X}_{images}[j]$ is the $j$th image in the input image series; $E_s'$ is the the raw output embedding corresponding to [CLS] through the transformer; $f_s$ is the projection head that maps the output embeddings $E_s'$ to a specified dimension $E_s \in R^D$.

**Text encoding:** The text encoder is to encode a text into a raw text embedding that is also projected to a specified dimension by the projection head, denoted as

$$E_t' = \text{Enc}_t(\mathbf{X}_{text}); E_t = f_t(E_t') \qquad (2)$$

where $\text{Enc}_t$ is the text encoder; $\mathbf{X}_{text}$ is the input text; $f_t$ is the projection head that maps the raw output embeddings $E_t'$ to a specified dimension $E_t \in R^D$. $E_t$ should have the same dimension with $E_s$ for contrastive learning. The text encoder is also used to encode high-level text summaries.

**High-level encoding:** The high-level encoding process involves feeding the image series embedding $E_s$ and the text embedding $E_t$ into an information extraction network to obtain their corresponding level 1 embeddings $E_s^{(1)}$ and $E_t^{(1)}$. These level 1 embeddings are subsequently passed into another

information extraction network to derive level 2 embeddings $E_s^{(2)}$ and $E_t^{(2)}$. In our specific implementation, the information extraction networks are structured as a multi-layer perceptron (MLP). By aligning these higher-level embeddings with the corresponding high-level text summaries, we aim to imbue these embeddings with more abstract and comprehensive high-level information. This enriched representation is valuable for enhancing the performance of downstream classification tasks. The high-level encoding process is formulated as

$$E_s^{(1)} = \text{MLP}_1(E_s); E_s^{(2)} = \text{MLP}_2(E_s^{(1)}); \qquad E_t^{(1)} = \text{MLP}_1(E_t); E_t^{(2)} = \text{MLP}_2(E_t^{(1)}) \quad (3)$$

## 3.2 HIGHER-LEVEL SUMMARY CONSTRUCTION

A sample could have multiple summaries in a level. In the case of our pre-training on the MIMIC-CXR dataset, we construct the high-level summaries by the provided 14 labels, including 12 specific diseases, *support devices* and *no finding*. Notably, we categorize *no finding* as a level 2 label, as it serves as a summary indicating the absence of any diseases. The remaining 13 labels are considered as level 1 labels.

For each label, we construct three distinct prompts to represent the negative, positive, and uncertain states of that label. For example, for label *Atelectasis*, the 3 prompts would be '*Disease Atelectasis is not found.*', '*Disease Atelectasis is found.*', '*Not sure if Disease Atelectasis is found.*'. Detail prompts for other labels are found in Appendix Table 4. These label prompts serve as the high-level summaries for high-level alignment. Each label prompt is input to the text encoder to get an embedding specific to that prompt.

The higher-level label encoding process is formulated as

$$\begin{aligned}
nl_i, pl_i, ul_i &= \text{Prompt}(X_{labels}[i]); \\
E'_{nl_i} &= \text{Enc}_t(nl_i); E'_{pl_i} = \text{Enc}_t(pl_i), E'_{ul_i} = \text{Enc}_t(ul_i) \\
E_{nl_i} &= f_l(E'_{nl_i}); E_{pl_i} = f_l(E'_{pl_i}); E_{ul_i} = f_l(E'_{ul_i})
\end{aligned} \quad (4)$$

where $X_{labels}[i]$ is the $i$th label of the input sample; $\text{Prompt}(\cdot)$ constructs 3 prompts for the label, representing the negative, positive, and uncertain states of that label. Each of these prompt is encoded into an embedding by the text encoder; $f_l$ is a projection head to map the embedding to a specified dimension. Notably, the embedding dimension for level 1 should match that of $E_s^{(1)}$, while the level 2 embedding dimension should be match that of $E_s^{(2)}$ for contrastive learning.

## 3.3 CASCLIP PRE-TRAINING

To pre-train a casCLIP , we set the training loss function a combination of two parts: the general CLIP loss function is used for the original image series-text alignment and the SAOD alignment loss for higher-level alignment, denoted as

$$L = L_{\text{CLIP}} + L_{\text{SAOD}} \quad (5)$$

**CLIP loss:** In the context of CLIP, given a batch of $N$ samples, it computes an $N \times N$ similarity matrix between image series and texts, where the diagonal elements are for paired image series and texts. CLIP loss is to maximize the diagonal elements in the similarity matrix in both row and column directions by a cross entropy loss function. The general CLIP loss is computed as

$$\begin{aligned}
S_{ij} &= E_{s_i} \cdot E_{t_j}^T / \tau_0; \\
L_{\text{CE}_i}^{\text{image}} &= -\log \frac{\exp(S_{ii})}{\sum_{j=1}^N \exp(S_{ij})}; L_{\text{CE}_i}^{\text{text}} = -\log \frac{\exp(S_{ii})}{\sum_{j=1}^N \exp(S_{ji})}; \\
L_{\text{CLIP}} &= \frac{1}{2N} \sum_{i=1}^N (L_{\text{CE}_i}^{\text{image}} + L_{\text{CE}_i}^{\text{text}})
\end{aligned} \quad (6)$$

where $\tau_0$ a learnable temperature parameter.

**Support Alignment Opposition De-alignment (SAOD):** SAOD is employed for higher-level alignment, where a single higher-level text summary may be associated with multiple image series, and

conversely, an image series may have multiple labels, each corresponding to various text summaries. In our design, each label is linked to three label prompt embeddings, representing the negative, positive, and uncertain status of that label. For a given sample with a specific label status, SAOD serves to maximize the similarity between the higher-level embeddings and the label prompt embedding of that particular status. Simultaneously, it seeks to minimize the similarity between the higher-level embeddings and the label prompt embeddings corresponding to other statuses. For sample $i$ with label $l$ in status $m$ (0=negative, 1=positive, 2=uncertain), we implement the SAOD alignment for level $k$ alignment as follows

$$S_{il}^{\text{image}} = [E_{s_i}^{(k)} \cdot E_{nl}^T, E_{s_i}^{(k)} \cdot E_{pl}^T, E_{s_i}^{(k)} \cdot E_{ul}^T,]/\tau_k; \quad S_{il}^{\text{text}} = [E_{t_i}^{(k)} \cdot E_{nl}^T, E_{t_i}^{(k)} \cdot E_{pl}^T, E_{t_i}^{(k)} \cdot E_{ul}^T,]/\tau_k;$$

$$L_{\text{CE}_i}^{\text{image}} = -\log \frac{\exp(S_{il}^{\text{image}}[m])}{\sum_{j=1}^3 \exp(S_{il}^{\text{image}}[j])}; \quad L_{\text{CE}_i}^{\text{text}} = -\log \frac{\exp(S_{il}^{\text{text}}[m])}{\sum_{j=1}^3 \exp(S_{il}^{\text{text}}[j])};$$

$$L_{\text{SAOD}_{il}}^{(k)} = (L_{\text{CE}_i}^{\text{image}} + L_{\text{CE}_i}^{\text{text}})/2$$

$$(7)$$

where $\tau_k$ a learnable temperature parameter for level $k$; $E_{nl}, E_{pl}, E_{ul}$ are the prompt embeddings corresponding to negative, positive and uncertain status of label $l$. Consequently, the overall SAOD loss in a batch is

$$L_{\text{SAOD}} = \frac{1}{N} \sum_{i=1}^N \sum_k \sum_l L_{\text{SAOD}_{il}}^{(k)} \tag{8}$$

where $k$ traverses all the levels and $l$ iterates through all the labels with known status for sample $i$. This formulation enables the SAOD loss to be computed across all levels and labels within the batch, facilitating the alignment and contrastive learning objectives.

## 3.4 Inference

Given an image series, inference aims to determine the presence of a specific disease, such as Cardiomegaly. Figure 3 illustrates the inference process using casCLIP. Firstly, The image series is passed to the multi-level image series encoder in casCLIP to get the image series embeddings $E_s^{(1)}$ and $E_s^{(2)}$ for level 1 and level 2, respectively. The multi-level image series encoder includes the image series encoder and the information extraction network in casCLIP. Secondly, we pass the 3 prompts associated with the disease to the text encoder to obtain embeddings for prompts corresponding to negative, positive and uncertain status of the disease. Thirdly, we compute the similarity between image series embeddings and the prompts for each level. This similarity score reflects the likelihood of a match between the image series and the corresponding disease status. These similarity scores can then be processed using a softmax function to derive the related probabilities. Through this process, casCLIP allows for the assessment of the presence of a specific disease in an image series based on the similarity between the image series and the corresponding prompts.

## 4 Experiments

In this section we conduct experiments to validate the effectiveness of our proposed casCLIP framework for zero-shot classification and linear-probe evaluation.

### 4.1 datasets

#### 4.1.1 Dataset for pre-training:

We pre-train casCLIP on MIMIC-CXR dataset (Johnson et al., 2019; Johnson et al.). The MIMIC-CXR database is a large publicly available dataset of chest radiographs with free-text radiology reports. The dataset contains 377,110 images corresponding to 227,835 radiographic studies performed at the Beth Israel Deaconess Medical Center in Boston, MA. The dataset is de-identified and protected health information (PHI) has been removed. Their also provide an official split for training, validation and test and no overlap patients on the three sets. Our pre-training is on the training set that contains 368,960 images for 222,758 studies. We validate the model on the validation set

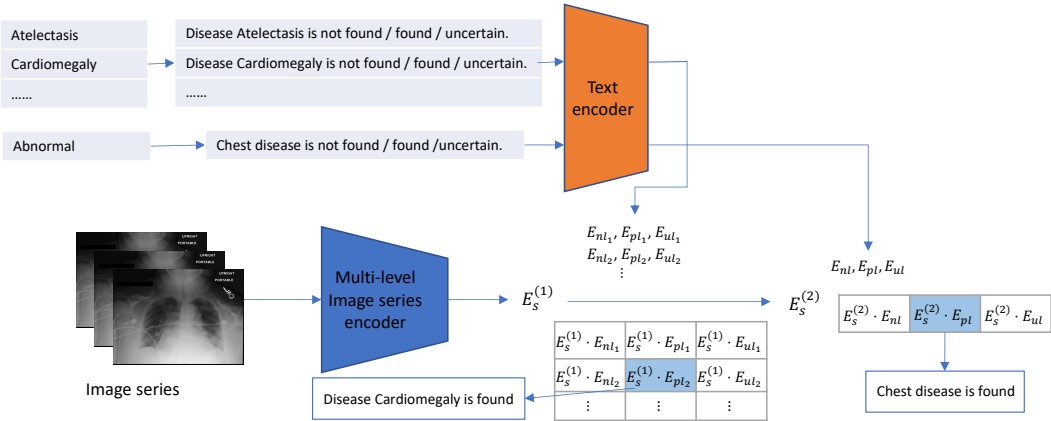

Figure 3: Inference with casCLIP.

that contains 2,991 images for 1,808 studies. The validation is during the pre-training process to save the model that perform best on validation set.

### 4.1.2 DATASET FOR DOWNSTREAM TASKS:

**MIMIC-CXR test set** is a split from MIMIC-CXR dataset. It contains 5,159 images for 3,269 studies.

**ChestX-ray14** (Wang et al., 2017) contains 112,120 frontal-view X-ray images of 30,805 unique patients, collected from the year of 1992 to 2015 by NIH(National Institutes of Health), with labels of 14 common diseases provided. This dataset is labeled per image, we consider each image as a study. We evaluate zero-shot classification on the provided test set that contains 25,596 images.

**CheXpert500.** CheXpert(Irvin et al., 2019) is a large dataset of chest X-rays and competition for automated chest x-ray interpretation, which features uncertainty labels and radiologist-labeled reference standard evaluation sets. We perform zero-shot evaluation on the collection of test set consisting of 500 studies from 500 patients. Eight board-certified radiologists individually annotated each of the studies in this test set following the same procedure and post-processing. We only evaluate 5 competition pathologies (i.e.,Atelectasis, Cardiomegaly, Consolidation, Edema, Pleural effusion) on this dataset, following the previous study (Tiu et al., 2022).

### 4.2 IMPLEMENTATION DETAILS

We benchmarked on the Mask R-CNN (He et al., 2017) with SwinTransformer (Liu et al., 2021) followed by a feature pyramid networks (FPN) (Lin et al., 2017) as the image encoder. We used the BERT model as the backbone of text encoder. The image encoder and text encoder are initialized with the GLIP-T (Li et al., 2022a) model. The original text embedding dimension output from text encoder is 768. The original image series embedding dimension output from image series encoder is 1280 and is then reduced to 768 by a linear projection head. The level 1 and level 2 embeddings are 128 and 64, respectively. The transformer to aggregate the image embeddings consists 2 layers of Multihead Attention. All The learnable temperature $\tau_k$ is initialized to 0.07. The training batch size is 32 and the max epoch is 30. we use AdamW optimizer (Loshchilov & Hutter, 2019) with both initial weight decay and initial learning rate euqal to 0.0001 during pre-training.

We implemented 4 variants for casCLIP. casCLIP_SAOD_h2 was implemented with 2 higher levels with SAOD alignment. casCLIP_h2 was implemented with 2 higher levels without SAOD alignment. casCLIP_SAOD_h1 was implemented with 1 higher levels with SAOD alignment. casCLIP_h1 was implemented with 1 higher levels without SAOD alignment.

Table 1: results on MIMIC-CXR dataset

|  | AUC | ACC | F1 |
|---|---|---|---|
| casCLIP_SAOD_h2 | **0.8735** | **0.8505** | **0.5897** |
| casCLIP_h2 | 0.8543 | 0.8367 | 0.5587 |
| casCLIP_SAOD_h1 | 0.8547 | 0.8489 | 0.5714 |
| casCLIP_h1 | 0.8549 | 0.8367 | 0.5641 |
| MedKLIP (Wu et al., 2023) | 0.6337 | 0.4776 | 0.3235 |

Table 2: results on ChestXray14 dataset

|  | AUC | ACC | F1 |
|---|---|---|---|
| casCLIP_SAOD_h2 | **0.7743** | **0.9160** | 0.2926 |
| casCLIP_h2 | 0.7249 | 0.8844 | 0.2868 |
| casCLIP_SAOD_h1 | 0.7673 | 0.8837 | **0.3294** |
| casCLIP_h1 | 0.7280 | 0.5728 | 0.2133 |
| MedKLIP (Wu et al., 2023) | 0.7134 | 0.7898 | 0.2465 |
| CheXzero (Tiu et al., 2022) | 0.7296 | 0.8278 | 0.2141 |
| BioViL (Boecking et al., 2022) | 0.6912 | 0.7916 | 0.1931 |
| GLoRIA (Huang et al., 2021a) | 0.6610 | 0.7700 | 0.1732 |
| ConVIRT (Zhang et al., 2022) | 0.6101 | 0.7102 | 0.1628 |

### 4.3 ZERO-SHOT CLASSIFICATION

We conduct zero-shot classification evaluation on 3 datasets: CheXpert, Chest-Xray14. We illustrate the results in Table 1, 2 and 3 for the 3 datasets respectively. Some results are from the related papers. From the results, our casCLIP can outperform other baselines.

### 4.4 ABLATION STUDY

In this section, we provide additional ablation studies on influence factors including the designed levels and the SAOD alignment. Comparing the results between casCLIP_SAOD_h2 and casCLIP_SAOD_h1, casCLIP_SAOD_h2 perform better than casCLIP_SAOD_h1, demonstrating 2 level hierarchical structure is better. casCLIP_SAOD_h2 performs better than casCLIP_h2, demonstrating the efficacy of SAOD alignment.

## 5 CONCLUSION

In conclusion, the development and exploration of the cascaded CLIP (casCLIP) model represent a significant leap forward in the field of medical image understanding. By harnessing the power of contrastive learning across multiple modalities, including medical images, textual reports, and

Table 3: results on CheXpert dataset

|  | AUC | ACC | F1 |
|---|---|---|---|
| casCLIP_SAOD_h2 | 0.8978 | **0.8640** | 0.6697 |
| casCLIP_h2 | 0.8912 | 0.8420 | 0.6428 |
| casCLIP_SAOD_h1 | 0.9002 | 0.8536 | 0.6584 |
| casCLIP_h1 | **0.9050** | 0.8620 | **0.6793** |
| MedKLIP (Wu et al., 2023) | 0.7871 | 0.7480 | 0.5112 |
| MedCLIP (Wang et al., 2022) | 0.8524 | 0.8360 | 0.6304 |
| CheXzero (Tiu et al., 2022) | 0.8890 |  | 0.6060 |
| GLoRIA (Huang et al., 2021a) |  | 0.6100 | 0.6700 |

hierarchical labels, casCLIP excels at capturing the rich and complex information present in the medical domain.

The model's pretraining on chest X-ray images with associated reports and high-level disease information extraction empowers it with the capability to effectively bridge the gap between visual and textual data in the medical field. Furthermore, casCLIP's adaptability to the multilevel of medical data positions it as a promising asset for medical diagnosis, treatment planning, and image-text retrieval tasks. Its multidimensional understanding of medical images and reports can contribute to more accurate and efficient healthcare practices.

As with any groundbreaking research, there remain opportunities and challenges on the horizon. Future work may focus on further refining casCLIP's performance, addressing potential biases in medical data, and ensuring its ethical use in healthcare. Nevertheless, the advent of casCLIP marks a significant advancement in leveraging state-of-the-art AI techniques for enhancing medical image understanding and holds great promise for the future of healthcare applications.

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

## A    APPENDIX

Table 4: The constructed label prompts in MIMIC-CXR dataset for high-level text summary.

|  | labels | status | prompts |
|---|---|---|---|
| level 1 | Atelectasis | negative | Disease Atelectasis is not found. |
| | | positive | Disease Atelectasis is found. |
| | | uncertain | Not sure if Disease Atelectasis is found. |
| | Cardiomegaly | negative | Disease Cardiomegaly is not found. |
| | | positive | Disease Cardiomegaly is found. |
| | | uncertain | Not sure if Disease Cardiomegaly is found. |
| | Consolidation | negative | Disease Consolidation is not found. |
| | | positive | Disease Consolidation is found. |
| | | uncertain | Not sure if Disease Consolidation is found. |
| | Edema | negative | Disease Edema is not found. |
| | | positive | Disease Edema is found. |
| | | uncertain | Not sure if Disease Edema is found. |
| | Enlarged Cardiomediastinum | negative | Disease Enlarged Cardiomediastinum is not found. |
| | | positive | Disease Enlarged Cardiomediastinum is found. |
| | | uncertain | Not sure if Disease Enlarged Cardiomediastinum is found. |
| | Fracture | negative | Disease Fracture is not found. |
| | | positive | Disease Fracture is found. |
| | | uncertain | Not sure if Disease Fracture is found. |
| | Lung Lesion | negative | Disease Lung Lesion is not found. |
| | | positive | Disease Lung Lesion is found. |
| | | uncertain | Not sure if Disease Lung Lesion is found. |
| | Lung Opacity | negative | Disease Lung Opacity is not found. |
| | | positive | Disease Lung Opacity is found. |
| | | uncertain | Not sure if Disease Lung Opacity is found. |
| | Pleural Effusion | negative | Disease Pleural Effusion is not found. |
| | | positive | Disease Pleural Effusion is found. |
| | | uncertain | Not sure if Disease Pleural Effusion is found. |
| | Pleural Other | negative | Pleural disease other than Effusion is not found. |
| | | positive | Pleural disease other than Effusion is found. |
| | | uncertain | Not sure if Pleural disease other than Effusion is found. |
| | Pneumonia | negative | Disease Pneumonia is not found. |
| | | positive | Disease Pneumonia is found. |
| | | uncertain | Not sure if Disease Pneumonia is found. |
| | Pneumothorax | negative | Disease Pneumothorax is not found. |
| | | positive | Disease Pneumothorax is found. |
| | | uncertain | Not sure if Disease Pneumothorax is found. |
| | Support Devices | negative | Support Device is not found. |
| | | positive | Support Device is found. |
| | | uncertain | Not sure if Support Device is found. |
| level 2 | No Finding | negative | Chest Disease is found. |
| | | positive | Chest Disease is not found. |
| | | uncertain | Not sure if Chest Disease is found. |

