# OpenReview forum: "Cascaded Contrastive Medical Language-Image Pretraining on Radiology Images"
_ICLR.cc/2024/Conference — ICLR 2024 Conference Withdrawn Submission_

### Official Review · Reviewer_V3gc · 2023-11-03

**Soundness:** 2 fair
**Presentation:** 3 good
**Contribution:** 2 fair
**Rating:** 3
**Confidence:** 3

**Summary:**

The authors propose a cascaded CLIP approach, which generates multiple image embeddings for multiple text levels and cascades the alignments between the text embeddings and the image embeddings instead of aligning an image embedding with multiple text embeddings simultaneously. They treat all images associated with a report as one entity that is fed into a multi-layer transformer to produce a single image embedding for all related images, which is then aligned with the report embedding. They also create an opposite
text summary for every text summary to increase the dissimilarity between the opposite text summary and the image, additionally to aligning the embeddings for the support text summary and the image embedding. They pretrain their proposed method (called casCLIP) on a public dataset and evaluate it on multiple other datasets.

**Strengths:**

- Very relevant study and application to use medical record information exhaustingly, which is particularly important as medical data is usually limited in size.
- The paper is written clearly and easy to follow, the main idea is well-motivated and the background section is adequate. The figures are well structured to explain the concept of the method.
- The method is evaluated on multiple openly available datasets and compared to relevant other approaches.
- An ablation study is performed to motivate some design choices of the method.

**Weaknesses:**

- No error bars or standard deviations are reported  - does that mean the experiments were run only once? Were the models tested for their sensitivity to initialization? This would limit the conclusions that can be drawn from the results.

- I could not find any links to any code release or repo - do the author intend to release the code and trained models? If so, please include the corresponding link in the paper. If not, I fear it’s not easy to reimplement the model by only reading the paper, based on the implementation details given in the text.

Minor remarks:
- In the last line of page 3 it says “aapproach”
- “Each prompt is feed” in caption of Fig. 2
- A space too many in Sec 3.3 at “To pre-train a casCLIP ,”
-In section 4.2 “All The learnable temperature”, as well as capitalize following sentence and correct “euqal”


----------------------------------------------------------------------------
Update after Author-Reviewer Discussion Phase

I acknowledge having read through the other reviewers' reviews who pointed out several serious concerns regarding the experimental setup which I had initially missed and are very important. I therefor lower my initial score.

**Questions:**

Why was MedKlip chosen as a competitor for the tests on MIMIC-CXR dataset? CheXzero seems to be better performing on the ChestXray14 dataset and the CheXpert dataset.

---

### Official Review · Reviewer_6KtK · 2023-11-07

**Soundness:** 2 fair
**Presentation:** 3 good
**Contribution:** 2 fair
**Rating:** 5
**Confidence:** 4

**Summary:**

This paper proposes an approach to improve CLIP models by leveraging labels with hierarchical structure in the medical domain. The paper contributes by introducing a novel architecture able to consider sequences of radiology images and align them with representations of textual information of different hierarchies. Support Alignment Opposition De-alignment (SAOD) is introduced to enhance alignment of hierarchical information. The zero-shot classification performance of the proposed methodology is evaluated on two additional datasets containing chest x-rays.

**Strengths:**

* (S1) The proposed methodology excels in its simplicity and is seemingly easy to implement.
* (S2) The method is general and applicable to any data where it is desirable to align image representations with hierarchical information.
* (S3) The paper is well-motivated.
* (S4) The introduction and methodology sections are well-written.
* (S5) The evaluation includes comparisons to recent SOTA CLIP models for biomedical data.
* (S6) The paper contains informative graphs, well-illustrating the proposed methodology.

**Weaknesses:**

* (W1) The experimental setup has baselines that may not be a fair comparison and the source of the reported baseline results are unclear. For instance ConVIRT and GLoRIA were originally pre-trained on proprietary datasets substantially different from MIMIC-CXR and none of the baseline methods use the same initialisation from GLIP-T as the proposed methodology.
* (W2) The ablation study does not investigate the impact of key parts of the proposed methodology. In particular it is unclear
	* how the model would perform without using hierarchical labels of any level,
	* the impact of using a sequence of radiology images instead of aligning the representation to each image,
	* what the impact of the GLIP initialization is compared to the more standard CLIP initialization.
* (W3) The paper is not well-situated in the literature. The authors note that other CLIP models have explored utilizing hierarchical information, however does not elaborate on how the proposed methodology is different from the mentioned papers. Further, the authors state that "few studies have been done on CLIP to handle multilevel medical information" but does not refer to any such studies.
* (W4) The proposed methodology is only applied to multilevel labels of limited sophistication, where the level 2 information simply indicates the existence of a chest disease. In particular, the method is not demonstrated on data where the labels are a hierarchy of diagnosis codes. For instance the ICD10 disease code corresponding to Atelectasis, mentioned in Figure 2, is a subclass of the more general disease code Pulmonary collapse. It would be beneficial to understand how the proposed methodology performs with hierarchies with a depth greater than two.
* (W5) It is not clear from section 4.2 exactly what architecture is used as the image encoder.
* (W6) It is not clear how `casCLIP_h1` and `casCLIP_h2` are implemented without SAOD alignment.
* (W7) The explanation of the SAOD alignment is overly complex. The paper would be more easier to understand if the fact that the loss is practically a repeat application of the CLIP loss, but on all possible labels of level $k$ instead of only those that appear in the batch.
* (W8) The discussions of the experiments and ablation studies are minimal:
	* the paper simply concludes that the proposed methodology outperform other baselines and that two-level information with SAOD alignment is the best.
	* the chosen baseline models are not discussed even though they differ substantially from the proposed method.
* (W9) It is not trivial how multiple levels of information is used in the downstream evaluation, since these datasets do not naturally contain hierarchical labels, however no further explanation is given. Section 3.4 states that inference is independent of the hierarchy, so i assume that only one level is used for the down-stream evaluation. A very interesting and central discussion is then why using multiple levels in the pre-training improve the alignment of the representation when only using one level in the downstream evaluation, however the paper does not include any such discussion.
* (W10) It is not clear from the paper how the multilevel information is extracted from the radiology reports. The pre-training dataset, MIMIC-CXR, contains a semi structured free-text radiology report suggesting that the extraction is rule-based, however no details are included in the paper.
* (W11) The proposed methodology is only demonstrated on datasets of limited diversity. An important application of models trained with hierarchical labels is the zero-shot generalization to data with unseen low level labels, but with high level labels similar to what appeared in the pre-training dataset. However, the proposed methodology is not demonstrated on any such dataset. An example of such a dataset could be COVIDx CXR-2 (Pavlova et al., 2022), which contains x-ray images of Pneumonia, also present in the MIMIC-CXR dataset, caused by either Covid or not.

References:

Maya Pavlova, Naomi Terhljan, Audrey G Chung, Andy Zhao, Siddharth Surana, Hossein Aboutalebi, Hayden Gunraj, Ali Sabri, Amer Alaref, and Alexander Wong. Covid-net cxr-2: An enhanced deep convolutional neural network design for detection of covid-19 cases from chest x-ray images. Frontiers in Medicine, 9, 2022.

**Questions:**

1. How does the model perform compared to the baseline models if no hierarchical labels are used?
2. How does the model perform compared to the baseline models if only single images are considered?
3. Did the authors pre-train any of the baseline models themself on MIMIC-CXR or did they use results from other papers different from the original articles where the models where trained on MIMIC-CXR? In the latter case, what are the source of the results?
4. Which architecture was used for the image encoder in the final implementation?
5. How are the labels acquired from the radiology report?
6. How does the models `casCLIP_h1` and `casCLIP_h2` utilize hierarchical labels if SAOD is not used?
7. See (W9) above. Why does the authors think using multiple levels in the pre-training improve the alignment of the representation when only using one level in the downstream evaluation?

Further, i would suggest improving the literature review to better understand the proposed methodology compared to other CLIP models using hierarchical information.

---

### Official Review · Reviewer_WmZb · 2023-11-08

**Soundness:** 1 poor
**Presentation:** 1 poor
**Contribution:** 1 poor
**Rating:** 3
**Confidence:** 4

**Summary:**

The paper propose a cascaded CLIP model which is able to overcome multilevel medical information, such as low-level and high-level text information, as well as input that are series of images rather than single images. The proposed method outperforms other baseline methods for classification of disease in Xray images.

**Strengths:**

Good introduction which motivates the need to develop a model capable of handling both series of image data and multi-level textual information.

**Weaknesses:**

The author suggests adding two consecutive networks on top of the existing text-image clip embedding to create two new "higher level" embeddings. These embeddings are trained via a "Support Alignment Opposition De-alignment Method" to better align the embeddings with 14 labels of interest (the 14 disease of the training dataset). The steps involved in this are:
1) Turn each label into a string: "Fracture":True becomes "Patient has a fracture" (positive sentence) and "Fracture": False becomes "Patient has no fracture" (negative sentence).
2) Have the higher level network predict a high level embedding from the image embedding
3) Calculate the correlation between the predicted high level embedding and the negative and positive sentence embeddings.
4) Apply softmax on the two correlations and train the network using cross-entropy.

This method seems very similar to training a network head for ordinary multi-class classification. The only difference seems to be that the article does the training in text-embedding-space instead of label-space. This is not problematic by itself, however, since test datasets contain the exact same 14 labels as the training dataset, I would argue that testing on these datasets is not "zero-shot". For this to be zero-shot, the test-datasets should contain different classes than the training dataset. It is therefore not possible to do a fair comparison with other zero-shot methods.


One of the stated main contributions of the paper is to allow for series of data rather than single images. However, the impact of this contribution is never evaluated, and it is not clear if the datasets even allow for this evaluation.

The metrics for MedKLIP in Table 1 and 2 are incorrect. The AUC and Accuracy of MedKLIP on the ChextXray14 dataset is stated in this paper to be 0.71 and 0.79, respectively, however, in the manuscript [Wu et al., 2023] the authors report 0.77 and 0.86, respectively.
Furthermore, MedKLIP used the MIMIC-CXR dataset to pre-train their model, and thus does not report any metrics for this dataset. How can this paper report metrics then?


The ablation study is unclear. The _h2 vs _h1 comparison appear to evaluate the impact of the multi-level input. If that is true, that it is not clear how this is possible without SAOD alignment.


The manuscript contains several typos, incomplete references, and odd formulation. One example is on page 7, "CheXpert is a large dataset of chest X-rays and competition for automated chest x-ray interpretation [...]".

**Questions:**

See weaknesses.

---

### Official Review · Reviewer_cZSY · 2023-11-10

**Soundness:** 2 fair
**Presentation:** 2 fair
**Contribution:** 3 good
**Rating:** 5
**Confidence:** 4

**Summary:**

This paper proposes a new way of classifying images. Since there are different requirements for information granularity in different clinical stages the authors argue that it is beneficial to have a model that can output different levels of details. By cascading methods that extract information on a different granularity level the authors create a method that is able to predict the disease accuracy more accurately compared to previous methods. The alignment between different hierarchies is done by using a method called support alignment opposition de-alignment (SAOD).

**Strengths:**

- This paper solves a real problem in the medical imaging domain. By considering information on different levels of granularity the problem is approached in a similar way as a clinician would analyse the problem, enabling the potential for integration in clinical workflow.
- The description of the training and inference procedure is described in detail and seems reproducible based on the information given in the paper alone.
- The method design which allows for pre-training with large image-text datasets, and inference on small image-only datasets is smart and practical.

**Weaknesses:**

- The writing of the paper is not clear, especially in the second half of the paper. It is not clear what the authors see as the most important contribution. The four contributions listed on page 2 do not represent the content of the paper and not are not equally supported with experimental results.
- caption of fig 2 is too long. Part of this information should have been integrated in the text. Captions of fig 3 and tab 1-3 are not informative enough.
- The reason for using image series instead of singular images is not motivated well enough and the benefit of this is not shown in the experiments.
- A comparison to chest X-ray classification methods that do not incorporate text in their pre-training could have been beneficial to further show the effect of this method,
-The construction of ablation studies is not clear. If SAOD is not used, does this mean that the information extraction network for that hierarchy is simply removed from the architecture?

Minor

- the notation of casCLIP_SAOD_h2 in text and tables makes the text less readable. This is especially apparent in section 4.4

- "their also provide .." page 6 - grammatical

- "the validation is .." page 7 -incorrect sentence construction

- "All The learnable temperature" page 7 - typo

- "As with any groundbreaking research,there remain opportunities and challenges on the horizon." page 9 - this is quite a strong statement, without a lot of content. Could be omitted.

- more work on the presentation of the paper is needed. Think about the size of text in figures and the locations of tables in the paper.

**Questions:**

1 How does this model perform if single images are used instead of image-series?

2 Generally Chest X-ray datasets only contain only for a part of the dataset the multiple X-ray views required for the image-series method. If only one view is available, does the method work with only one image or is the data of this patient omitted from the dataset.

3 is it possible to add a third hierarchy level to this model? how would this look like and what would be the challenges (methodological and data-wise?)